# Unsupervised Domain Adaptation of Brain MRI Skull Stripping Trained on Adult Data to Newborns: Combining Synthetic Data with Domain Invariant Features

**Abbas Omidi**[1,2]                                        ABBAS.OMIDI@UCALGARY.CA
[1] *Electrical and Software Engineering, University of Calgary, Calgary, AB, Canada*
[2] *Hotchkiss Brain Institute, University of Calgary, Calgary, AB, Canada*

**Amirmohammad Shamaei**[1,2]                            AMIRMOHAMMAD.SHAMAEI@UCALGARY.CA

**Anouk Verschuur**[3,4,5]                                ANOUK.VERSCHUUR@UCALGARY.CA
[3] *Department of Radiology, Isala Hospital Zwolle, The Netherlands*
[4] *Image Sciences Institute, University Medical Center Utrecht, The Netherlands*
[5] *Department of Pediatrics, Section of Neonatology, University of Calgary, Canada*

**Regan King**[5,6]                                        REGAN.KING@UCALGARY.CA
[6] *Alberta Children's Hospital Research Institute, Calgary, AB, Canada*

**Lara Leijser**[2,5,6]                                    LARA.LEIJSER@UCALGARY.CA

**Roberto Souza**[1,2]                                     ROBERTO.SOUZA2@UCALGARY.CA

**Editors:** Under Review for MIDL 2024

## Abstract

Skull-stripping constitutes a crucial initial step in neuroimaging analysis, and supervised deep-learning models have demonstrated considerable success in automating this task. However, a notable challenge is the limited availability of publicly accessible newborn brain MRI datasets. Furthermore, these datasets frequently use diverse post-processing techniques to improve image quality, which may not be consistently feasible in all clinical settings. Additionally, manual segmentation of newborn brain MR images is labor-intensive and demands specialized expertise, rendering it inefficient. While various adult brain MRI datasets with skull-stripping masks are publicly available, applying supervised models trained on these datasets directly to newborns poses a challenge due to domain shift. We propose a methodology that combines domain adversarial models to learn domain-invariant features between newborn and adult data, along with the integration of synthetic data generated using a Gaussian Mixture Model (GMM) as well as data augmentation procedures. The GMM method facilitates the creation of synthetic brain MR images, ensuring a diverse and representative input from multiple domains within our source dataset during model training. The data augmentation procedures were tailored to make the adult MRI data distribution closer to the newborn data distribution. Our results yielded an overall Dice coefficient of $0.9308 \pm 0.0297$ ($mean \pm std$), outperforming all compared unsupervised domain adaptation models and surpassing some supervised techniques previously trained on newborn data. This project's code and trained models' weights are publicly available at https://github.com/abbasomidi77/GMM-Enhanced-DAUnet

**Keywords:** Skull-stripping, Domain Adaptation, Deep Learning, GMM, MRI

## 1. Introduction

Advancements in neuroimaging, particularly through Brain Magnetic Resonance Imaging (MRI), have revolutionized diagnostic approaches in neurology. MRI now has become a vital diagnostic and monitoring tool for neurological illnesses since it is a thorough, non-invasive imaging technology, that aids in the understanding of various neurologic conditions (Combes et al., 2022). A fundamental step in MRI analysis is skull-stripping, the process of isolating brain tissue from the rest of the head image, which is critical for accurate diagnosis and effective research in neurology (Rehman et al., 2020).

Newborn brain MRI presents distinctive challenges due to the unique physiological attributes of the neonatal brain, such as differences in brain structure and the contrast between white matter (WM) and gray matter (GM) due to incomplete myelination (Dubois et al., 2014). Also, motion artifacts and additional structures, such as the neck and portions of the shoulders, show up in newborn MR images. These differences result in a substantial domain shift between adult and newborn MRI data, making it challenging to apply segmentation models trained on adult data to newborn data. The limited availability of annotated newborn MRI datasets further hampers the training of supervised deep-learning models.

This paper introduces a new unsupervised method for newborn brain MRI skull-stripping that combines domain adversarial approaches to learn domain invariant features (Ganin et al., 2016) with a Gaussian Mixture Model (GMM) to generate synthetic data (Billot et al., 2023), and a data augmentation strategy to help reduce the distribution shift between adult and newborn MRI data. This combination enables our system to accurately segment newborn brain images despite this domain's inherent challenges and without the need for any labelled newborn MR images. A detailed ablation study and comparison against state-of-the-art supervised models highlight the advantages of the proposed method.

## 2. Related work

Domain adaption (DA) techniques are essential for reliable and generalizable model performance on a variety of datasets in the field of medical image segmentation, especially MRI skull-stripping (Ghafoorian et al., 2017; Zhong et al., 2021). These methods tackle the issue of domain shift, which can hinder the transferability of models between domains because of variations in image acquisition protocols (Kondrateva et al., 2021; Full et al., 2021).

A prominent DA approach is adversarial DA (Ganin and Lempitsky, 2015; Tzeng et al., 2017; Dinsdale et al., 2021). These methods revolve around fostering the creation of domain-invariant features and closing the gap between different domains. It leverages a discriminator to explicitly distinguish between domains, encouraging the segmentation network to learn domain-invariant features and showcasing the potential to yield more flexible and effective models. However, its efficiency decreases when there are complex domain shifts.

Classic methods such as BET (Brain Extraction Tool) (Smith, 2002), BSE (Brain Surface Extractor) (Shattuck et al., 2001), and Robex (Iglesias et al., 2011) have laid the groundwork for skull-stripping, demonstrating effectiveness in various contexts yet facing limitations in adaptability and precision, particularly with heterogeneous data like neonatal MRI scans.

In skull-stripping, automated methods have evolved from traditional image-processing techniques to deep learning-based methods. Techniques like BEaST (Eskildsen et al., 2012), region growing (Lu et al., 2003), and deformable models (Colliot et al., 2006) were pioneering

but often fell short due to their reliance on predefined features and parameters, making them less adaptable to the variability inherent in MRI scans.

Recent advancements in deep learning and the development of infant-specific methods, such as SSCNN (Jog et al., 2019), iBEAT (Dai et al., 2013), and the approaches proposed by Lucena et al. (Lucena et al., 2019) and Chen et al. (Chen et al., 2018), have aimed to tailor computational techniques to the distinct anatomical and contrast characteristics of neonatal and infant brains. These methods represent innovative strides toward more accurate brain imaging analyses in neonates and infants. However, despite their advancements, these methods still face challenges related to domain shifts, data scarcity or limited availability, and the need for extensive labeled datasets for training. This is particularly problematic given the high variability present in real-world clinical settings. Moreover, the utilization of comprehensive datasets, such as the developing Human Connectome Project (dHCP) dataset, underscores the critical need for robust and diverse data to effectively train and validate these deep learning approaches (Edwards et al., 2022). However, the dHCP dataset is limited by its single-center nature and the use of sophisticated post-processing to enhance image quality, practices not universally feasible in clinical environments. These factors contribute to potential generalization issues across different centers. Hence, while invaluable, relying solely on dHCP for developing skull-stripping models might not ensure broad applicability in diverse clinical settings.

To mitigate these issues, recent research has explored the use of DA techniques (Wang and Deng, 2018; Munk et al., 2023; Farahani et al., 2021; Ben-David et al., 2006). For instance, Billot et al. introduced a novel method utilizing a GMM to create synthetic brain MRI scans of various contrasts and resolutions (Billot et al., 2023). These synthetic scans are then used to train segmentation models, showing promising results in handling domain shifts. However, this data-centric strategy addresses the domain shift challenge predominantly through the lens of data availability and diversity. The architecture of a model is pivotal in determining its learning efficiency and generalization power across varying domains. If the architectural aspects of the model remain unchanged or unoptimized, the benefits of a richer and more diverse dataset may not be fully realized.

Our previous work addresses the challenge of skull-stripping in newborn MRI data by developing an architecture trained on adult MRI data for use on newborns (Omidi et al., 2024). While this approach represents an improvement in handling the domain shift between adult and newborn data without supervision, it faces constraints due to limited source data. This discrepancy highlights a critical issue in medical imaging: the balance between innovative model architecture and the availability of diverse, representative training data.

## 3. Method

### 3.1. Gaussian Mixture Model

Building upon the domain randomization strategy introduced by Billot et al., we utilize a generative model to produce a sequence of diverse synthetic brain MRI scans featuring significant anatomical and intensity variations. By fully randomizing the generation parameters, the synthetic training data encompasses a wide range of contrasts and resolutions, enabling the segmentation network to generalize without retraining.

In the Billot et al. approach, the generative model assumes the availability of $N$ training label maps $\{S_n\}_{n=1}^N$ defined over discrete spatial coordinates $(x, y, z)$ at high resolution $r_{\mathrm{HR}}$. These label maps, taking values from a set of $K$ labels, can be obtained manually,

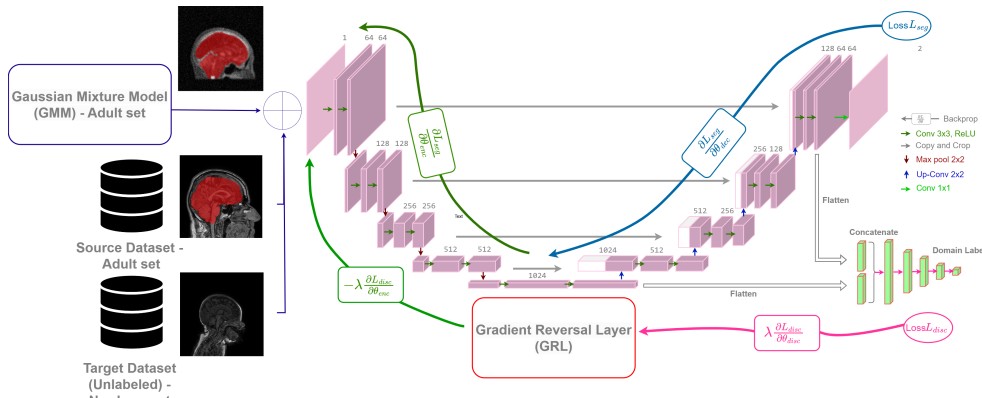

Figure 1: Overview of the method. Synthetic data generated by the GMM block and the Calgary-Campinas adult dataset are the source data used to learn the U-net's parameters (upper arrows). All data, including the unlabelled newborn images, are used to learn the domain classifier parameters and encourage the U-net's encoder to learn domain invariant features across source and target domains (lower arrows) using a gradient reversal strategy.

automatically, or through a combination of both methods, as long as they share the same labeling convention.

An initial high-resolution synthetic scan $G$ is then generated by sampling from a Gaussian Mixture Model (GMM) conditioned on the deformed label map $L$. The means and standard deviations of the GMM denoted as $\mu_G = \{\mu_k\}_{k=1}^K$ and $\Sigma_G = \{\sigma_k\}_{k=1}^K$, respectively, are sampled at each mini-batch from normal distributions to randomize the contrast of $G$. The standard deviations $\Sigma_G$ jointly model tissue heterogeneities and the thermal noise of the scanner. The synthetic scan $G$ is formed by independently sampling at each location $(x, y, z)$ from the normal distribution indexed by $L(x, y, z)$, with mean $\mu_{L(x,y,z)}$ and standard deviation $\sigma_{L(x,y,z)}$.

During each training iteration, a segmentation $S_i$ is randomly selected from the training dataset and applied a set of deformations, including affine, elastic, and motion transformations, to enhance the variability of the available segmentations. The method produces two volumes: a synthetic image $G$ sampled from the generative model and its corresponding segmentation target $S$. The segmentation target $S$ is obtained by taking the deformed label map $L$ and resetting the label values of background structures to zero, resulting in a map with $K' \leq K$ labels.

By exposing the segmentation network to a different combination of contrast, resolution, morphology, artifacts, and noise at each mini-batch, the aim is to achieve robust segmentation of brain MRI scans across a wide range of contrasts and resolutions without the need for retraining. Diverging from Billot et al. methodology, where training images are generated on the fly, we generate training images in advance to streamline the training phase.

## 3.2. Domain Adversarial Neural Network (DANN)

Our segmentation framework is built upon a 3D U-Net model (Figure 1). In our DA model, we interweave three principal alterations into the classical U-Net construct. To distinguish between MRI brain images of adults (source) and newborns (target), we incorporated a discriminator network. This network takes as input the features extracted from the last

layer of the decoder and the bottleneck features. This process is applied to both source sets (GMM-Enhanced Dataset and Calgary-Campinas Dataset) and our target dataset (Newborn). This discriminator plays a pivotal role in classifying the domain of input images, leveraging both high and low-level features.

To encourage the generation of features independent of the imaging domain, we implemented a gradient reversal layer, inspired by the work of Ganin et al. (Ganin and Lempitsky, 2015). This layer adjusts backpropagation gradients inversely through a scaling factor, providing a strategic means to promote domain-invariant feature extraction. Also, we combined the output from the encoder's bottleneck with the final layer before the discriminator. The operational mechanism of our network is shown through the following equation:

$$\Delta\boldsymbol{\theta} = -\mu\left(\frac{\partial\mathcal{L}\mathrm{seg}}{\partial\boldsymbol{\theta}\mathrm{dec}} + \lambda\frac{\partial\mathcal{L}\mathrm{disc}}{\partial\boldsymbol{\theta}\mathrm{disc}} - \frac{\partial\mathcal{L}\mathrm{seg}}{\partial\boldsymbol{\theta}\mathrm{enc}} + \lambda\frac{\partial\mathcal{L}\mathrm{disc}}{\partial\boldsymbol{\theta}\mathrm{enc}}\right) \tag{1}$$

The provided equation illustrates the update process of network parameters, denoted as $\Delta\boldsymbol{\theta}$, during the learning phase, guided by the learning rate ($\mu$) and scaling factor ($\lambda$). The equation indicates that the decoder parameters ($\boldsymbol{\theta}_{\mathrm{dec}}$) undergo updates based solely on the segmentation loss ($\mathcal{L}_{\mathrm{seg}}$), the discriminator parameters ($\boldsymbol{\theta}_{\mathrm{disc}}$) undergo updates based solely on the discriminator loss ($\mathcal{L}_{\mathrm{disc}}$), while the encoder parameters ($\boldsymbol{\theta}_{\mathrm{enc}}$) undergo updates considering both the segmentation loss ($\mathcal{L}_{\mathrm{seg}}$) and the discriminator loss ($\mathcal{L}_{\mathrm{disc}}$).

### 3.3. Data Augmentation

Addressing the unique challenges in newborn brain MRI, we tackled the issue of not fully myelinated GM and WM tissue contrast by inverting the WM-GM contrast in the Calgary-Campinas dataset. This modification, aimed at aligning with the inverted contrast observed in newborns, facilitated more effective skull-stripping by training the model on a mix of original and contrast-inverted images. The contrast inversion was implemented as follows:

$$\bar{C}(i) = \max_{j \in M}[C(j)] - C(i), \forall i \in M \tag{2}$$

where $C(i)$ is the original voxel intensity and $\bar{C}(i)$ the transformed intensity for voxel $i$ within the WM-GM mask, with $M$ being the set of WM-GM voxels.

To mitigate the effects of motion artifacts due to newborn movements during scans, we adapted our GMM-Enhanced dataset with motion artifacts and Gaussian Blur transformations. These adaptations, aimed at mimicking newborn MRI conditions, employed the MONAI framework's blur function (Cardoso et al., 2022) and motion artifacts protocol proposed by Zaitsev et al. (Zaitsev et al., 2015), enhancing the dataset's suitability for newborn imaging analysis.

### 3.4. Loss Function

In our model, we use the Dice loss function for segmentation, and for the discriminator's performance evaluation, we rely on Binary Cross-Entropy (BCE) loss. The composite loss function can be mathematically expressed as follows:

$$\mathcal{L}_{\mathrm{total}} = \sum_{i=1}^{N_1}[1 - \mathrm{Dice}(\hat{\mathbf{y}}_i, \mathbf{y}_i) + \mathrm{BCE}(\mathbf{u}_i, \mathbf{v}_i)] + \sum_{j=1}^{N_2}\mathrm{BCE}(\mathbf{u}_j, \mathbf{v}_j) \tag{3}$$

Here, $\mathcal{L}_{\text{total}}$ denotes the overall loss. The first part computes the Dice Loss for the predicted segmentation ($\hat{\mathbf{y}}_i$) against the ground truth ($\mathbf{y}_i$) for $N_1$ source images. The second part calculates the BCE Loss for these source images ($\mathbf{u}_i$) against their domain labels ($\mathbf{v}_i$). The equation also includes BCE Loss for $N_2$ target images ($\mathbf{u}_j$) and their domain labels ($\mathbf{v}_j$). This part is crucial for enabling the model to adjust to the target domain's characteristics.

## 4. Experiments

### 4.1. Datasets

#### 4.1.1. Calgary-Campinas Dataset

In our research, we employed the Calgary-Campinas public brain MRI dataset, comprising 359 T1-weighted, 3D, 1 mm isotropic adult brain MRIs, with a gender distribution of 176 males and 183 females (Souza et al., 2018). This dataset has vital features like skull-stripped brain masks and segmentation masks for WM and GM generated using FSL software (Jenkinson et al., 2012). The images were acquired using MRI scanners from various manufacturers and magnetic field strengths.

#### 4.1.2. GMM-Enhanced Dataset

The GMM-enhanced dataset comprises 420 synthetic 3D samples generated from 20 adult brain segmentation masks (Billot et al., 2023). The number of synthetic samples was obtained empirically. The GMM approach allowed us to create a diverse and comprehensive set of data without relying on actual newborn MRI scans.

#### 4.1.3. Newborn Dataset

Our study incorporated a private dataset from the Alberta Children's Hospital, gathered using a GE 3T MRI scanner. This dataset consists of 12 high-resolution T1-weighted, 3D newborn brain MRIs (7 females and 5 males), each with dimensions of 1 mm $\times$ 1 mm $\times$ 0.5 mm. For evaluation purposes, brain masks were manually acquired on ten sagittal slices, each spaced ten slices apart, from five out of the twelve samples in the test dataset.

### 4.2. Training Details

Our model was trained for 500 epochs with a batch size of four on an A-100 GPU. Each epoch took around 20 minutes to complete. For training, we employed a strategy of extracting $96 \times 96 \times 96$ patches, which was executed using the MONAI framework (Cardoso et al., 2022). The patch-based method not only managed the data volume effectively but also optimized our resource usage. Furthermore, the datasets are provided to the network in a balanced fashion, indicating that we duplicated the limited target dataset multiple times to align its quantity with that of our source datasets.

To evaluate and compare the performance of the models, we used the Dice coefficient and the 95th percentile Hausdorff distance metrics. These metrics were calculated on the newborn test set. Details of the data distribution across training, validation, and testing phases are provided in Table 1.

Table 1: Summary of the datasets used in the experiments.

| Data | Train Set | Validation Set | Test Set |
|---|---|---|---|
| Adult Data | 243 | 116 | 0 |
| Newborn Data | 5 | 2 | 5 |
| GMM-Enhanced Data | 400 | 20 | 0 |

### 4.3. Ablation Study and Model Comparison

To evaluate the effectiveness of our proposed approach, we conducted three ablation studies focusing on different aspects of the proposed model. We also compared the proposed model against the U-Net model trained on the Calgary-Campinas dataset, HippoDeep (Thyreau et al., 2018), SynthStrip (Billot et al., 2023), and our previous work (Omidi et al., 2024). Both HippoDeep and SynthStrip had access to labelled newborn data during training.

In the first ablation study, we utilized a basic U-Net model and trained it with the Calgary-Campinas and GMM-Enhanced datasets (Named: U-Net GMM in Table 2). The aim was to see how much better the U-Net segmentation would be when exposed to the GMM-Enahnced data during training.

The second ablation study added the domain adversarial component to our model so it would learn domain invariant features (Named: DA GMM in Table 2). This model also used GMM-Enhanced data during training, but the GMM-Enhanced data used did not leverage our data augmentation strategy (see Section 3.3). This study aimed to quantify the impact of the DA component in our proposed network architecture.

In the third ablation study, the proposed data augmentation strategy was added to our training data (Named: Proposed Method in Table 2). This study evaluated the impact of the proposed data augmentation on the segmentation results.

## 5. Results

The Dice coefficient and the $95^{th}$ percentile Hausdorff distance metrics are summarized in Table 2. Our proposed method achieved the best performance among unsupervised models in the newborn test set. It demonstrated a 1.67% improvement in terms of the mean values of the Dice coefficient and reduced standard deviation by approximately 8.3% when compared to the second-best unsupervised method, represented by Omidi et al. Additionally, in the Hausdorff distance results, our method remained the top performer among unsupervised approaches, showing a 6.4% improvement in the Dice coefficient values and a reduction of variation by approximately 20.9% when compared to the second-best unsupervised model.

Furthermore, when compared to state-of-the-art supervised models, our model outperformed the Hippodeep model by 0.93% in the Dice coefficient and 9.8% in the Hausdorff distance. Additionally, our model exhibits a slight deviation of only 1.11% less than Synth-

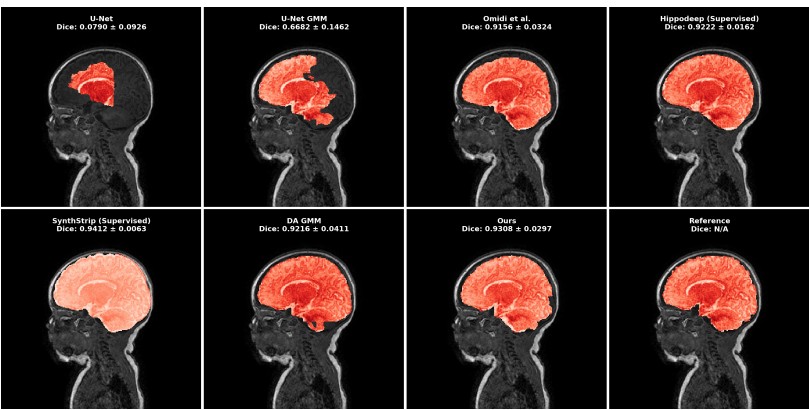

Figure 2: Representative illustration of the results for the different methods compared.

Strip in the Dice coefficient and 8.6% for the Haussdorff distance metric. For a visual representation of the segmentation masks that reflect the overall results of the newborn test set, please refer to Figure 2. For a discussion on the limitations of our study, we have included a failure case analysis in the Supplementary Material.

Table 2: Summary of results for the newborn MRI test set. Best results are shown in bold.

| Approach | Hausdorff Distance | Dice Coefficient |
|---|---|---|
| U-Net | 8.9220 ± 0.5335 | 0.0790 ± 0.0926 |
| U-Net GMM | 5.0056 ± 0.6523 | 0.6682 ± 0.1462 |
| Omidi et al. | 3.6621 ± 0.4572 | 0.9156 ± 0.0324 |
| Hippodeep* | 3.7995 ± 0.2343 | 0.9222 ± 0.0162 |
| SynthStrip* | **3.1570 ± 0.1389** | **0.9412 ± 0.0063** |
| DA GMM | 3.4528 ± 0.2053 | 0.9216 ± 0.0411 |
| Proposed Method | 3.4272 ± 0.3614 | 0.9308 ± 0.0297 |

*Supervised models which have been trained on including newborn data.

## 6. Discussion

The ablation studies showed that combining synthetic data for training with architectural model improvements specifically designed for DA purposes and a data augmentation strategy tailored to close the gap between the adult and newborn data distribution shift resulted in better segmentation results. The GMM-enhanced data improved the Dice coefficient from 0.08 (U-Net) to 0.67 (U-Net GMM). While the DA component of our model further improved the Dice results to 0.92 (DA GMM). Finally, our data augmentation component that added to our training set elements, such as motion corruption and WM-GM contrast inversion, further improved our Dice coefficient to 0.93. The results of these ablation studies highlight the importance of combined approaches that look both at data aspects and model architecture for medical image segmentation problems.

Also, the generalizability of the proposed method to other real-world scenarios may be impacted by the limited availability of publicly accessible newborn brain MRI datasets, as the model's performance has been validated on a small sample of the newborn MRI data landscape. In medical imaging, the diversity of data, encompassing variations in imaging protocols and patient demographics, plays a crucial role in the robustness of models. Without access to a broader range of datasets, there's an inherent risk that the model may not capture the full spectrum of variability in newborn brain anatomy and pathology.

## 7. Conclusion

Our proposed unsupervised skull-stripping method for newborn MRI achieved results comparable to supervised methods. The proposed method leverages DA techniques, GMM-generated synthetic data, and a data augmentation approach tailored to newborns to overcome the issue of domain shift across adult and newborn MRI data.

Our results showed a 1.67% improvement in the Dice coefficient compared to the best existing unsupervised methods in the literature. Also, when compared to state-of-the-art supervised models, our approach surpasses the Hippodeep model by 0.93% in the Dice coefficient and closely trails SynthStrip by only 1.11%. It is worth noting that SynthStrip is trained on a range of newborn MRIs, and Hippodeep utilizes a dataset of over 5000 MRI scans. Our results pave the way for developing new deep learning DA segmentation models that operate both from the data and model architecture perspectives.

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

## Appendix A. Supplementary material

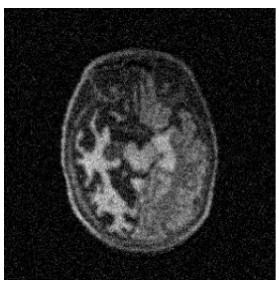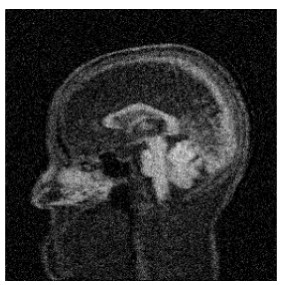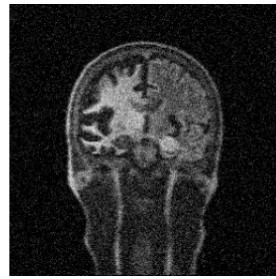

Figure 3: Instances of synthetically generated MRI data using the GMM.

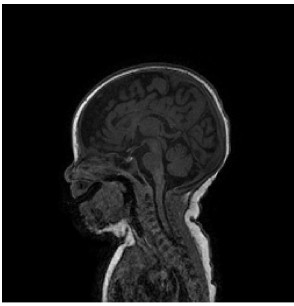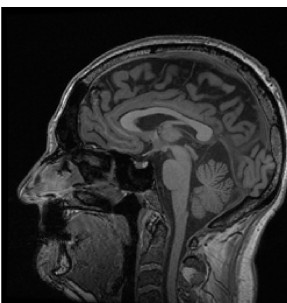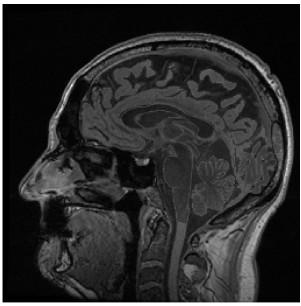

Figure 4: (Left) T1-weighted brain MRI of a newborn subject. (Centre) T1-weighted brain MRI of an adult subject. (Right) T1-weighted brain MRI of an adult subject after contrast inversion.

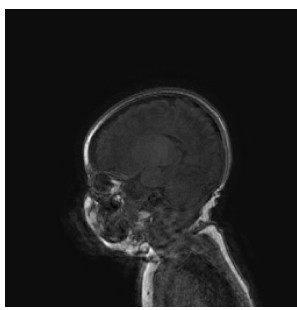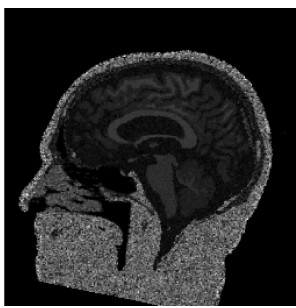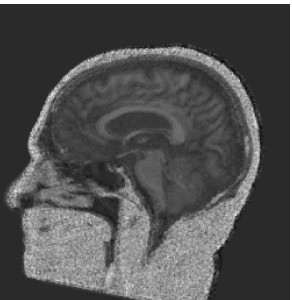

Figure 5: (Left) T1-weighted brain MRI of a newborn subject. (Centre) Synthetic T1-weighted brain MRI of an adult subject. (Right) T1-weighted brain MRI of an adult subject after motion artifact transformation.

## Appendix B. Failure Case Analysis

Despite the demonstrated strengths of our method, it encounters a limitation in effectively performing skull-stripping operations on cases with atypical skull structures. For instance, as depicted in Figure 6, our method struggles with accurately annotating the brain of an infant whose skull is unusually elongated. While this challenge is not unique to our approach and all other unsupervised methods similarly struggle, it is noteworthy that supervised methods exhibit superior performance in this area. This could be attributed to the exposure of supervised methods to newborn brain imagery, enabling them to better recognize and adapt to the variations in infants' brain shapes and sizes. Nonetheless, it is important to highlight that, despite this specific shortcoming, our method still marks a considerable advancement over our previous study (Omidi et al., 2024).

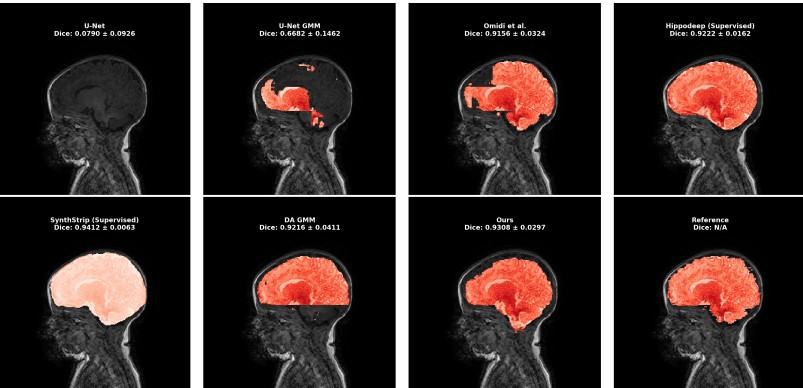

Figure 6: A Failure case of our method and comparison with other methods.

Additionally, our methodology faces challenges with input images where brain regions are exceptionally dark with very low contrast between the brain and skull, as can be seen in Figure 7. This often stems from the inherent difficulties of scanning newborns. Such conditions necessitate the development and incorporation of adaptive domain methods that can mitigate the effects of these scanning irregularities, enhancing the model's ability to accurately identify and process brain areas even under less-than-ideal conditions.

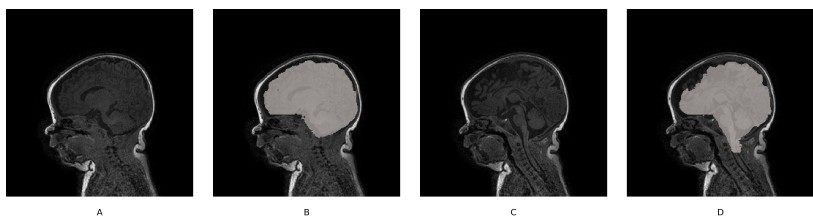

Figure 7: (A) T1-weighted brain MRI of a newborn subject. (B) Our method's skull-stripping result on the subject (A). (C) Same subject as (A) but in a different slice. (D) Our method's skull-stripping result on the image slice shown in (C) partially failed.

