# OpenReview forum: "Unsupervised Domain Adaptation of Brain MRI Skull Stripping Trained on Adult Data to Newborns: Combining Synthetic Data with Domain Invariant Features"
_MIDL.io/2024/Conference — MIDL 2024 Poster_

### Official Review · Reviewer_qxAN · 2024-02-22

**Confidence:** 4
**Preliminary Rating:** 4
**Recommendation:** Poster
**Final Rating:** 4

**Summary:**

The paper proposes a methodology that combines domain adversarial models, synthetic data generation using a Gaussian Mixture Model (GMM), and data augmentation procedures for unsupervised domain adaptation of brain MRI skull stripping from adult data to newborns. The proposed methodology addresses the challenge of domain shift when applying supervised models trained on adult brain MRI datasets to newborns.

**Strengths:**

- The paper addresses the challenge of domain shift when applying supervised models trained on adult brain MRI datasets to newborns, which is a significant issue in neuroimaging analysis.

- The proposed methodology combines domain adversarial models, synthetic data generation using a Gaussian Mixture Model (GMM), and data augmentation procedures, which collectively contribute to achieving accurate skull stripping in newborn brain MR images

- The availability of the project's code and trained models' weights on a public repository promotes reproducibility and further research in the field of newborn brain MRI analysis

.

**Weaknesses:**

- The paper does not provide a detailed discussion or analysis of the limitations or potential drawbacks of the proposed methodology.

- The scarcity of publicly accessible newborn brain MRI datasets with manually annotated masks is acknowledged as a challenge. Still, the paper does not discuss this limitation's potential impact on the proposed method's generalizability and applicability.

- The paper does not provide a comprehensive comparison with other state-of-the-art unsupervised domain adaptation models or discuss the specific limitations of those models that the proposed method overcomes.

**Detailed Comments:**

- Best to provide a detailed discussion or analysis of the limitations or potential drawbacks of the proposed methodology.

- The scarcity of publicly accessible newborn brain MRI datasets with manually annotated masks is acknowledged as a challenge. So, it is necessary to discuss this limitation's potential impact on the proposed method's generalizability and applicability.

- Please have a comprehensive comparison with other state-of-the-art unsupervised domain adaptation models or discuss the specific limitations of those models that the proposed method overcomes.

- Please give discuss of the potential limitations or challenges associated with the use of synthetic data generated using a Gaussian Mixture Model (GMM) and the impact of this synthetic data on the accuracy and reliability of the skull stripping results.

**Justification Of Final Rating:**

This study introduces a method that combines domain adversarial models to learn domain-invariant features between newborn and adult brain MRI datasets. It also integrates synthetic data generated using a Gaussian Mixture Model and data augmentation techniques. The results of this study outperform unsupervised domain adaptation models and even some supervised techniques previously trained on newborn data. The authors have satisfactorily addressed all concerns.

**Justification Of The Preliminary Rating:**

The proposed method opens up possibilities for developing new deep learning domain adaptation segmentation models that can operate effectively on both adult and newborn MRI data, contributing to advancements in the field.

**Questions To Address In The Rebuttal:**

1. What are the limitations of the proposed methodology?

2. How does the scarcity of annotated datasets impact the generalizability of the method?

3. What are the specific limitations of other unsupervised domain adaptation models?

4. What role does the Gaussian Mixture Model (GMM) play in generating synthetic brain MR images?

**Special Issue:**

No

---

> ### Author Response · Authors · 2024-03-17
> **Our response**
>
> We thank the reviewer for their relevant and detailed feedback and appreciate the opportunity to discuss the limitations and challenges of our proposed methodology further. Please see our answers to your questions below.
>
> **Comment:** What are the limitations of the proposed methodology?
>
> **Response:** In response to the request for a discussion on the situations in which our method works poorly, we have included a section devoted to examining the shortcomings and failure scenarios in the Supplementary Material. For instance, our method struggles with infant subjects whose skull is unusually elongated, and in general, it encounters limitations in cases with atypical skull structures. Additionally, our methodology faces challenges with input images where brain regions are exceptionally dark and with very low contrast.
>
> **Comment:** How does the scarcity of annotated datasets impact the generalizability of the method?
>
> **Response:** To address this concern comprehensively, we've included more details in our paper's discussion section, to discuss this limitation's potential impact on the proposed method's generalizability and applicability. In medical imaging, the diversity of data, encompassing variations in imaging protocols and patient demographics, plays a crucial role in the robustness of machine learning models. Without access to a broader range of datasets, there's an inherent risk that the model may not capture the full spectrum of variability in newborn brain anatomy and pathology. As a result, we agree that the generalizability of our proposed method to other datasets and real-world scenarios may be impacted by the scarcity of publicly accessible newborn brain MRI datasets, as the model's performance has been validated on a relatively small sample of the newborn MRI data landscape.
>
> **Comment:** What are the specific limitations of other unsupervised domain adaptation models?
>
> **Response:** Our exploration into unsupervised domain adaptation for newborn brain MRI data highlighted a significant gap in the literature, as there are no existing unsupervised models directly addressing this challenging application. This absence underscored the innovative aspect of our research, which applied data augmentation strategies (white-matter and gray-matter contrast inversion and motion artifacts) to make the adult data distribution more similar to newborn data. Given the availability of supervised models trained on newborn data (SynthStrip is trained on newborns labeled MRI from ages 0 to 18 months, and Hippodeep uses over 5000 MRI samples), we opted to use them as baselines for comparisons. Our method outperformed one of these models, demonstrating its efficacy and contributing a novel solution to the field. As future work, we agree that other domain adaptation techniques could be adapted/implemented for the specific problem of skull-stripping newborn brain MRI.
>
> **Comment:** What role does the Gaussian Mixture Model (GMM) play in generating synthetic brain MR images?
>
> **Response:** To give a fuller description of how the GMM approach is applied in sample synthesis, we have expanded on Section 3.1. This provides a more thorough explanation of the GMM's function in our methodology and how it's integrated into the creation of synthetic data.
>
> We hope these clarifications and reflections address your concerns.

---

> > ### Comment · Reviewer_qxAN · 2024-03-26
> >
> > The authors have addressed all of my concerns. I have no more comments.

---

### Official Review · Reviewer_ycv6 · 2024-02-28

**Confidence:** 5
**Preliminary Rating:** 1
**Final Rating:** 2

**Summary:**

The authors propose a new framework to carry out skullstripping in newborn brain MRI. Their solution combines domain adversarial models learning domain-invariant features between newborn and adult data, along with the integration of GMM-generated synthetic data and data augmentation procedures.
This formulation is a modification of an existing tool and the motivation for it is the lack of available newborn neuroimaging data set.

The experiments are run on a fairly small data set and result in satisfactory skullstripping results.

The authors committed to sharing their code on GitHub.

**Strengths:**

A full pipeline for skullstripping was proposed that could overcome the obstacle of very limited data availability.
The authors committed to sharing their code on GitHub.
The submission is clearly written.

**Weaknesses:**

The technical novelty is low and also the motivation cited is no longer valid (given the availability newborn neuroimaging data with including skullstripping mask).

The motivation / background section lacks some key citations regarding prior work.

No definition of the desired skull-stripped result is provided.

The generated motion artifacts (used for data augmentation) seem to be suboptimal (if the Appendix figure is a representative example)

**Detailed Comments:**

One key description I miss in this submission is the lack of a definition of the skull-stripping mask here. Knowing whether the training data did or did not include external CSF is very crucial when evaluating and comparing to other methods. SynthStrip, for example, includes CSF in the foreground. Does the new tool?

The motivation of the tool development is weak due to abundant (compared to the data set used by the authors) labeled newborn data availability. The best example would be the publicly available dHCP data cohort that is publicly distributed with postprocessing outcomes (including skull stripping and volumetric segmentation). The current test data set is also very small with 10 slices manually annotated from 5 subjects.

The background section lacks some classic methods, such as BET, BSE, Robex. In particular, some more recent DL methods (EVAC+) and infant-specific ones (SSCNN, iBEAT, and dCHP).

In order to provide a wide variety of variability in the training data set, SynthSeg is used for data generation. It seems to me that the proposed scaling range [.9-1.2] cannot be sufficient to bridge the newborn-adult domains, given that the newborn brain often doubles in size within the first two years of life.

If I understand correctly, a discriminator network is defined by the authors to determine whether the input is from newborn or adult subjects; given the above-mentioned scaling difference, this should be a straightforward task.

The motion artifacts implementation seems suboptimal. If Fig 5 is representative, the motion artifact seems like a pediatric scan overlaid on the adult and not an adult motion artifact.

It is the Calgary-Campinas Dataset that is used for the experiments, along with a small data set from Alberta. As mentioned above, I do not understand why the dHCP cohort was not relied on. It is actually even bigger than the one used and has skull-tripped and segmented solutions shared.

The Alberta newborn data set is also used. For testing, 5 data sets are selected (where 10 slices from each are manually annotated). This seems to be too small when a newborn datasets is readily available.

Additionally
immature --> not fully myelinated
Table 2 comment: "trained on" newborn data --> "trained on including" newborn data -- This is a very important distinction!
"The main difficulty in performing a more detailed evaluation is the lack of public newborn brain MRI datasets." That used to be true but for the past several years it is not the case.

**Justification Of Final Rating:**

The motivation / background section has been updated regarding previous work and the definition of the desired skull-stripped result has been provided. The idea about learning from rich adult datasets is interesting, but I would argue that the availability of various publicly available newborn brain MRI datasets makes the experimental section weak with its limited test data.

I change my score to weak reject due to the updates made in the submission.

**Justification Of The Preliminary Rating:**

The motivation/background section for the proposed tool development is weak.
There is a lack of prior work citation.
There is a lack of precise task definition.
There is minimal novelty in the proposed new framework.

**Questions To Address In The Rebuttal:**

What is the reason for not looking for publicly available newborn data sets?
Does the scaling range in the data augmentation phase cover the variability in the newborn population?

---

> ### Author Response · Authors · 2024-03-17
> **Our response (Part 1)**
>
> Thank you for your valuable and detailed feedback, which has led us to clarify and enhance our manuscript in several key areas. Please see our answers to your comments below:
>
> **Comment:** One key description I miss in this submission is the lack of a definition of the skull-stripping mask here. Knowing whether the training data did or did not include external CSF is very crucial when evaluating and comparing to other methods. SynthStrip, for example, includes CSF in the foreground. Does the new tool?
>
> **Response:** We do include external CSF similar to SynthStrip. We clarified this in the methods section.
>
> **Comment:** The motivation of the tool development is weak due to abundant (compared to the data set used by the authors) labeled newborn data availability. The best example would be the publicly available dHCP data cohort publicly distributed with postprocessing outcomes (including skull stripping and volumetric segmentation). The current test data set is also very small with 10 slices manually annotated from 5 subjects.
>
> **Response:** We only partially agree with the reviewer’s comment. The dHCP study is a landmark study, and a reference was added to the manuscript. We agree that in future works, our model should also be evaluated using their dataset and others potentially available. However, the dHCP has limitations:
> - Data distribution is made through torrents, and seeds are often unavailable. Also, the high-speed networks of many universities block these types of downloads for cybersecurity reasons. To this date, we have not been able to download all anatomical scans and corresponding masks.
> - The dHCP data is single-centre. It was collected using a 3T Philips Achieva scanner located at the Evelina London Children’s Hospital. Training the skull-stripping model using only their data is unlikely to generalize well across centres. This has been explicitly shown in the case of skull-stripping in research by other groups [1].
> - The dHCP scans (T1w) are obtained twice and combined using sophisticated post-processing algorithms to improve image quality by mitigating artifacts like motion. Such high-quality data is not feasible in most clinical settings. Though we could have used dHCP when developing our method, it would not necessarily generalize well to new data due to potentially poorer data quality of images collected in other centres.
> - We agree that our current test set is small and should be increased in future works. Nevertheless, our opinion is that the proposed technique is novel. It combines synthetic data with learning domain-invariant features, and our results using a limited dataset show promise for future use. We clarified in the paper the pilot nature of this study.
>
> **Comment:** The background section lacks some classic methods, such as BET, BSE, Robex. In particular, some more recent DL methods (EVAC+) and infant-specific ones (SSCNN, iBEAT, and dCHP).
>
> **Response:** We agree with the reviewer, and we have expanded our background section to include a discussion on classic methods (BET, BSE, Robex) and recent deep learning and infant-specific methods, providing a more comprehensive overview of the field.

---

> > ### Author Response · Authors · 2024-03-17
> > **Our response (Part 2)**
> >
> > **Comment:** In order to provide a wide variety of variability in the training data set, SynthSeg is used for data generation. It seems to me that the proposed scaling range [.9-1.2] cannot be sufficient to bridge the newborn-adult domains, given that the newborn brain often doubles in size within the first two years of life.
> >
> > **Response:** The scaling range of [0.9-1.2] was specifically chosen not to replicate neonate brain sizes but to introduce a broader spectrum of adult data variability. We do not need to generate synthetic images that look precisely like newborns. Our discriminator, as we detail in our following response, encourages our model to learn invariant features across adult and newborn domains. Our ablation study also supports our implementation since adding the synthetic data and data augmentation improved our model, and the addition of the adversarial component further improved our results.
> >
> > **Comment:** If I understand correctly, a discriminator network is defined by the authors to determine whether the input is from newborn or adult subjects; given the above-mentioned scaling difference, this should be a straightforward task.
> >
> > **Response:** The discriminator seeks to distinguish between newborn and adult images. We agree with the reviewer that this is a trivial task. However, as we train our model, we “encourage” it to learn features that are not discriminative of these two groups (i.e., domain-invariant features), which is at the root of domain adversarial domain adaptation techniques [2].
> >
> > **Comment:** The motion artifacts implementation seems suboptimal. If Fig 5 is representative, the motion artifact seems like a pediatric scan overlaid on the adult and not an adult motion artifact.
> >
> > **Response:** Thank you for the comment. We updated the figure to show a less extreme case of motion artifacts. The motion artifacts were generated using existing methods [3].
> >
> > **Comment:** It is the Calgary-Campinas Dataset that is used for the experiments, along with a small data set from Alberta. As mentioned above, I do not understand why the dHCP cohort was not relied on. It is actually even bigger than the one used and has skull-tripped and segmented solutions shared.
> > The Alberta newborn data set is also used. For testing, 5 data sets are selected (where 10 slices from each are manually annotated). This seems to be too small when a newborn datasets is readily available.
> >
> > **Response:** Please see our answer to your second comment. We agree that a more detailed evaluation of the method should be performed in future works.
> >
> > **Comment:** Additionally immature --> not fully myelinated Table 2 comment: "trained on" newborn data --> "trained on including" newborn data -- This is a very important distinction! "The main difficulty in performing a more detailed evaluation is the lack of public newborn brain MRI datasets." That used to be true but for the past several years it is not the case.
> >
> > **Response:** Thank you for your comment. We fixed this in the paper and also removed the sentence about the difficulty in performing a more detailed evaluation due the lack of public newborn brain MRI datasets and rephrased it to indicate the pilot nature of our work.
> >
> > We hope to have addressed/mitigated the reviewer’s concerns and hope that they will reconsider our score.
> >
> > **[1]** Shirokikh, B., Zakazov, I., Chernyavskiy, A., Fedulova, I. and Belyaev, M., 2020. First U-Net layers contain more domain specific information than the last ones. In Domain Adaptation and Representation Transfer, and Distributed and Collaborative Learning: Second MICCAI Workshop, DART 2020, and First MICCAI Workshop, DCL 2020, Held in Conjunction with MICCAI 2020, Lima, Peru, October 4–8, 2020, Proceedings 2 (pp. 117-126). Springer International Publishing.
> >
> > **[2]** Ganin, Y., Ustinova, E., Ajakan, H., Germain, P., Larochelle, H., Laviolette, F., March, M. and Lempitsky, V., 2016. Domain-adversarial training of neural networks. Journal of machine learning research, 17(59), pp.1-35.
> >
> > **[3]** Zaitsev, M., Maclaren, J. and Herbst, M., 2015. Motion artifacts in MRI: A complex problem with many partial solutions. Journal of Magnetic Resonance Imaging, 42(4), pp.887-901.

---

> > ### Comment · Reviewer_ycv6 · 2024-03-18
> >
> > The dHCP download site moved to https://nda.nih.gov/. The download mechanism does not include torrents and seeds any more.

---

> > > ### Author Response · Authors · 2024-03-18
> > > **Our response**
> > >
> > > Thank you for pointing us to the new download mechanism. As stated in our responses, we plan to use the dHCP data in future works to evaluate our model more in-depth. Also, according to the dHCP release notes, this download mechanism started in February 2024 (https://biomedia.github.io/dHCP-release-notes/), which was after the 2024 MIDL paper registration deadline.

---

> > > > ### Comment · Reviewer_ycv6 · 2024-03-26
> > > >
> > > > Thanks for the authors' responses to my questions.
> > > >
> > > > I appreciate the clarification on the tools task and the addition to the background section about already existing tools.
> > > >
> > > > I think we will have some disagreement remaining about the availability of newborn datasets. I mentioned the dHCP cohort as an example, but more varied (obtained at various sites, on various scanners) have also been available for years. See, for example, NIH MRI Study of Normal Brain Development, UNC/UMN Baby Connectome Project. A more complete validation / experimental section could be established by the use of any of these.
> > > >
> > > > re motion mitigation. "We updated the figure to show a less extreme case of motion artifacts" The modified example in the figure looks more reasonable. I did not find a tool associated with [3] though, that would generate these artifacts. [3] describes a set of tools that can be used to mitigate motion artifacts.

---

### Official Review · Reviewer_SSqP · 2024-03-06

**Confidence:** 3
**Preliminary Rating:** 4
**Recommendation:** Poster
**Final Rating:** 4

**Summary:**

This paper tries to tackle the challenge of brain MRI skull stripping for newborns. The authors proposed a combination of adversarial learning and GMM-based data augmentation to encourage the model to learn domain-invariant features while leveraging the domain gap between rich labeled adult and insufficient unlabeled newborn samples. The overall quality of the manuscript is acceptable, and even though the major components of the proposed method are adopted from existing works, the experiments show some promising potential of their method compared to others.  I would recommend accepting if my concerns below could be addressed.

**Strengths:**

1. The strength is moderate. Although many of the major components are not original, the results still show improvements over existing works.
2. The structure and language are fine as well.
3. This paper is an update of their previous work, with improved performance.

**Weaknesses:**

1. Lacking sufficient details in Section 3.1 because this is one of the major components of the method; even though it is not original, details must be provided to offer readers a clearer concept
2. The ablation study needs improving; the proposed discriminator lacks further inspection.
3. No limitation analysis, which might be crucial for medical applications.

**Detailed Comments:**

1. Section 3.1 regarding the Gaussian Mixture Model needs to be elaborated a bit more, although it is introduced in another paper. It remains unclear how this GMM method is applied in synthesizing samples. Besides, the set of deformations applied during training is just a very common data augmentation strategy. There is no need to allocate many lines in its details.
2. Despite GMM and the proposed DA, the discriminator in the model also acts as a constraint to limit the learning of domain-related features. However, no ablation study was conducted to verify that.
3. I would expect some words discussing failure (or worse) results in comparison to the results from other methods and possible explanations.
4. In Section 6, the first two numbers should be 0.08 (0.0790) and 0.67 (0.6682).

**Justification Of Final Rating:**

The authors have successfully addressed all my concerns. The newly added explanation of GMM in Section 3.1 eases the understanding of data augmentation. The failure case analysis further enhances the overall organization. Several typos have been corrected.

**Justification Of The Preliminary Rating:**

I would recommend weak acceptance for the following reasons:
1. This paper offers a moderately good method for tackling skull stripping for newborn brain MRIs; the proposed method is original even though the novelty of each individual component is not original. The experiments show a certain level of improvement over other existing works.
2. Despite the improved performance, some explanations, like how the GMM works and the motivation for employing discriminator in the model, are missing.

**Questions To Address In The Rebuttal:**

Updated Section 3.1, the ablation studies, and failure case analysis.

**Special Issue:**

No

---

> ### Author Response · Authors · 2024-03-17
> **Our response**
>
> We appreciate the detailed feedback provided and have taken steps to address the concerns raised. Please see our answers to your comments below.
>
> **Comment:** Section 3.1 regarding the Gaussian Mixture Model needs to be elaborated a bit more, although it is introduced in another paper. It remains unclear how this GMM method is applied in synthesizing samples. Besides, the set of deformations applied during training is just a very common data augmentation strategy. There is no need to allocate many lines in its details.
>
> **Response:** We have elaborated on Section 3.1 to provide a clearer explanation of the GMM method's application in synthesizing samples. This includes a more detailed description of the GMM's role in our methodology and its integration into the synthetic data generation process. Also, since our code is publicly available, we minimized the focus on common data augmentation strategies and have thus condensed this section accordingly, as the reviewer suggested.
>
> **Comment:** Despite GMM and the proposed DA, the discriminator in the model also acts as a constraint to limit the learning of domain-related features. However, no ablation study was conducted to verify that.
>
> **Response:** We appreciate the opportunity to clarify the existence of an ablation study, which indeed examines the effect of the discriminator's absence on our model's performance. This study is detailed in Section 4.3 under the designation Unet GMM. This variant of our model utilizes the same data and architectural framework, excluding the discriminator component. This comparison underscores the discriminator's significant role in enhancing the model's ability to generalize across domains by mitigating the learning of domain-specific features. We clarified our text to remove any potential confusion by readers.
>
> **Comment:** I would expect some words discussing failure (or worse) results in comparison to the results from other methods and possible explanations.
>
> **Response:** Thank you for your comment. We added a section in the supplementary material that analyzes these failure cases.
>
> **Comment:** In Section 6, the first two numbers should be 0.08 (0.0790) and 0.67 (0.6682).
>
> **Response:** We have revised this issue. Thank you for pointing it out.
>
> We hope that these clarifications and reflections help address your concerns.

---

### Meta-Review · Area_Chair_MBeD · 2024-03-29

**Recommendation:** Accept (Poster)
**Confidence:** 5

**Metareview:**

While there is a point of contention in the actual usefulness of the *application* of the method given the availability of other datasets, the reviewers agree that there is value in the method itself, combining domain adaptation with synthetic data for enhanced DA. Moreover, the interactions between authors and reviewers have significantly improved the submission. I am recommending borderline acceptance for this paper.

---

### Decision · Program_Chairs · 2024-04-05

Accept (Poster)